# Association of Breastfeeding for the First Six Months of Life and Autism Spectrum Disorders: A National Multi-Center Study in China

**DOI:** 10.3390/nu14010045

**Published:** 2021-12-23

**Authors:** Saijun Huang, Xin Wang, Tao Sun, Hong Yu, Yanwei Liao, Muqing Cao, Li Cai, Xiuhong Li, Lizi Lin, Xi Su, Jin Jing

**Affiliations:** 1Department of Child Healthcare, Affiliated Foshan Maternity and Child Healthcare Hospital, Southern Medical University, Foshan 528000, China; tufanqie@163.com (S.H.); fsfyst2020@163.com (T.S.); yu376@163.com (H.Y.); liaoyanwei@tom.com (Y.L.); 2The Second School of Clinical Medicine, Southern Medical University, Guangzhou 510280, China; 3Department of Maternal and Child Health, School of Public Health, Sun Yat-sen University, Guangzhou 510080, China; wxin3@mail2.sysu.edu.cn (X.W.); caomq6@mail.sysu.edu.cn (M.C.); caili5@mail.sysu.edu.cn (L.C.); lixh@mail.sysu.edu.cn (X.L.); 4Guangdong Provincial Engineering Technology Research Center of Environmental Pollution and Health Risk Assessment, Department of Occupational and Environmental Health, School of Public Health, Sun Yat-sen University, Guangzhou 510080, China; linlz@mail.sysu.edu.cn

**Keywords:** autism spectrum disorders, breastfeeding, children, epidemiology

## Abstract

Previous studies have shown that exclusive breastfeeding is associated with lower odds of having autism spectrum disorders (ASD) in children, but data are lacking in Asian countries, especially China. This cross-sectional study of seven cities in China collected data from August 2016 to March 2017 from 6049 toddlers aged 16–30 months and their parents who responded to questionnaires. The breastfeeding status was collected via questionnaires based on recommendations from the World Health Organization. The standard procedure for screening and diagnosis was applied to identify toddlers with ASD. Among the 6049 toddlers (3364 boys [55.6%]; mean [SD] age, 22.7 [4.1] months), 71 toddlers (1.2%) were identified as ASD. The prevalence of exclusive breastfeeding, partial breastfeeding, and not breastfeeding was 48.8%, 42.2%, and 9.1%, respectively. Compared to toddlers with exclusive breastfeeding, toddlers with partial breastfeeding or without breastfeeding had higher odds of having ASD (odd ratios [OR]: 1.55, 95% confidence interval [CI]: 0.90–2.74; OR: 2.34, 95% CI: 1.10–4.82). We did not find significant modification of demographic characteristics on the associations. The results remained robust in multiple sensitivity analyses. Toddlers without breastfeeding for the first six months of life had higher odds of having ASD, and our findings shed light on the necessity of strengthening public health efforts to increase exclusive breastfeeding in China.

## 1. Introduction

The 2019 Global Burden of Disease study indicated that an estimated 28.3 million people were identified as having autism spectrum disorders (ASD) [1], characterized by deficits in social communication, repetitive behaviors, and highly restricted interests [2]. Although limited data of national prevalence of ASD in China have been estimated, a recent study has reported a prevalence of 0.70% in Chinese children aged 6–12 years [3]. Early intervention is seen as a priority, but the strength of evidence for most of the approaches remained uncertain [4]. Identifying modifiable factors in early life is still needed to tailor prevention strategies.

Breastfeeding promotes the achievement of early childhood milestones (language, cognition, fine motor skills, etc.) [5,6], and previous studies have identified the role of breastfeeding in neurodevelopment among newborns [7,8,9,10]. However, the associations between breastfeeding and ASD have remained unclear (Appendix A in Supplementary Materials) [11,12,13,14,15,16,17,18,19,20,21,22,23,24,25,26,27,28,29,30,31]. Although the latest meta-analyses of 13 studies provided evidence that breastfeeding may protect against ASD [32], several newly published studies were not included, providing additional information with inconsistent results. One cross-sectional study from Korea showed that breastfeeding was associated with higher odds of ASD compared to formula-feeding [31]. Two case-control studies (the US [25] and India [27]) and a birth cohort from Denmark [26] have indicated that a shorter breastfeeding duration is associated with higher risk of ASD in children. By contrast, one case-control study in the US observed null association between breastfeeding initiation and the risk of ASD using a standard diagnostic procedure with standard assessment [28]. Most of the studies were conducted in Western countries, and a few of them were carried out in Asian countries, especially in China. We only found one cross-sectional study in Shenzhen, China indicating that breastfeeding was associated with autistic traits in toddlers [29]. To our knowledge, no study has been conducted to investigate the association between breastfeeding and ASD with detailed information of the diagnostic processes in China. In addition, the Diagnostic and Statistical Manual 5th Edition (DSM-5) has replaced the previous DSM-IV and DSM-IV-Text Revision (DSM-IV-TR) diagnostic criteria of ASD, which were used in most previous studies. More studies are needed by considering the current diagnostic criteria.

Based on the above, we have conducted a nationwide cross-sectional study with the standard procedure of ASD screening and diagnosis, and we aim to investigate the associations between breastfeeding and the odds of ASD in Chinese toddlers. We hypothesized that toddlers without breastfeeding might have higher odds of being diagnosed with ASD.

## 2. Materials and Methods

### 2.1. Study Population and Overall Design

This cross-sectional study was embedded in our previous study of validating the Chinese version of the Modified Checklist for Autism in Toddlers, Revised with Follow-Up (M-CHAT-R/F) between August 2016 and March 2017. The details were described elsewhere [33]. This study adopted a convenient cluster-sampling strategy and generated a national sample of seven cities in six provinces from five geographical regions of China, namely, the Northern Region (Beijing City), the Western Region (Chongqing City and Guiyang City, Guizhou), the Southern Region (Guangzhou City and Foshan City, Guangdong Province), the Central Region (Wuhan City, Hubei Province), and the Eastern Region (Hangzhou City, Zhejiang Province). At each study site, we contacted the head of the regional maternal and child health hospital to obtain approval for the validation study. All hospital directors agreed to participate in the study. We requested that participants should come from a variety of sources, including hospital-based, community-based, and school-based populations. Finally, a total of 7928 toddlers aged 16–30 months were recruited from 7 tertiary hospitals, 21 communities, and 7 kindergartens, who agreed to join and complete the validation study. During the validation study, all 7928 caregivers of the toddlers were asked to participate in an additional survey, of which 6049 agreed to complete by questionnaires. The comparison of demographic characteristics of participants with or without the information of questionnaires is shown in Appendix A.

We followed the Strengthening the Reporting of Observational Studies in Epidemiology (STROBE) reporting guideline [34] for cross-sectional studies. This study was approved by the Ethical Review Committee for Biomedical Research, Sun Yat-sen University. All participants provided written informed consent and we informed all participants of the purpose of the study.

### 2.2. Procedure of ASD Screening and Diagnosis

The two-stage process was applied to confirm the ASD diagnosis of the included population. In the first stage, the M-CHAT-R/F was used for the screening. In the second stage, we conducted the standard diagnostic process if the screening was positive. Toddlers with positive screening further underwent a face-to-face Childhood Autism Rating Scale (CARS) performed by trained psychometrists and/or licensed psychologists in tertiary hospitals of the study area. The diagnostic process also included 30 min of parent interviews and interactions with toddlers. The final diagnosis was made according to the 30 min. interaction and the DSM-5 diagnostic criteria.

### 2.3. Covariates

Demographic information and other self-reported risk factors were collected via questionnaires, including the child’s age, sex, only child, maternal age, maternal education level, yearly household income, ethnic background, study area, preterm birth, and pregnancy information (complications and depression), and second-hand smoke exposure. We also collected information on self-reported weight and height before pregnancy, and we calculated the body mass index (BMI) as weight in kilograms divided by the square of height in meters. We then defined overweight and obese using Chinese adult references [35].

### 2.4. Breastfeeding

We administered a question of infant feeding to caregivers in the questionnaires. Caregivers were asked to report the extent of breastfeeding for the first six months according to the WHO breastfeeding definitions [36] (exclusive [i.e., only breastmilk or only breast milk and water given], partial [i.e., mixture of breast milk and formula given], or not breastfeeding [i.e., fully formula-fed]).

### 2.5. Statistical Analysis

Data were analyzed from 1 July to 13 September 2021. We calculated mean (SD) values for continuous variables and percentages for categorical variables. The *t*-test was used to explore continuous variables, and the *χ^2^* test was used to analyze the differences between toddlers with and without ASD.

The logit function was used to fit the generalized linear model to analyze the associations between feeding patterns for the first six months of life and the odds of ASD in toddlers. We fitted three models as follows: (1) crude model without adjustment; (2) adjusted Model One with covariates of demographic and socio-economic factors, including information on the child’s age, sex, only child, maternal age, maternal education level, yearly household income, ethnic background, and study area; (3) adjusted Model Two with further adjustments of autism-related risk factors, including information on preterm birth, pregnancy information (complications and depression during pregnancy), second-hand smoke exposure, and overweight/obesity before pregnancy.

Subgroup analysis stratified by demographic characteristics including information on sex, only child, maternal age, maternal education level, and yearly household income was conducted, and the differences between groups were tested by adding the interaction term. We did not conduct subgroup analyses for the ethnic background because the models failed to converge due to the small cases in the subgroups. We performed several sensitivity analyses: (1) we used Firth’s bias reduction method to fit the logistic regression model to minimize the analysis bias caused by small samples, rare events, and complete separation [37]; (2) we reanalyzed the data only using toddlers born at full-term, toddlers whose mothers had no complications and depression during pregnancy, and toddlers whose mothers were not overweight or obese before pregnancy.

Statistical software R (R Core Team 2019) version 3.6.1 was used for statistical analysis. We expressed the result as odd ratios with a 95% confidence interval. All applicable tests were bilateral tests, and *p* < 0.05 was statistically significant.

## 3. Results

### 3.1. Demographic Information and Maternal Risk Factors during Pregnancy

A total of 6049 toddlers with an average age of 22.7 months and their parents participated in the survey (Table 1). There were 3364 (55.6%) toddlers who were boys, and most of them were only children (70.7%). The prevalence of preterm birth was 8.4%, while most of the toddlers were of Han nationality (93.8%). The prevalence of complications and depression during pregnancy was 14.2% and 24.6% in the mothers, and 7.8% of them were overweight/obese before pregnancy. Of the mothers, 13.9% had been exposed to second-hand smoke. A comparison of toddlers with and without ASD is shown in Appendix A

### 3.2. Prevalence of ASD Associated with Breastfeeding

Among 6049 toddlers, 71 toddlers were diagnosed with ASD with a prevalence rate of 1.2%, and more boys (*n* = 62) were identified as having ASD (Figure 1). In the included toddlers, the prevalence of exclusive breastfeeding, partial breastfeeding and not breastfeeding were 48.8%, 42.2%, and 9.1%, respectively. In toddlers with ASD, partial breastfeeding (52.1%) was the most common breastfeeding status. Detailed data are shown in Appendix A

### 3.3. Associations of Breastfeeding in the First Six Months of Life with ASD among Toddlers

As shown in Table 2, we found that compared with toddlers with exclusive breastfeeding, toddlers without breastfeeding have higher odds of having ASD (OR = 2.19, 95% CI, 1.04–4.46) in the adjusted Model 1 by adjusting for demographic information. The results remained similar when we further adjusted for autism-related risk factors (OR = 2.34, 95% CI, 1.10–4.82). We did not observe associations between partial breastfeeding and ASD (OR = 1.55, 95% CI, 0.90–2.74).

### 3.4. Subgroup Analysis

As shown in Table 3, there was no statistical evidence of interactions between child sex, only child, maternal age, maternal education, yearly household income, and breastfeeding status on risk of ASD with *P*_interaction_ = 0.66, 0.75, 0.45, 0.79, and 0.42, respectively.

### 3.5. Sensitivity Analyses

The results remained similar when using Firth’s Bias-Reduced Logistic Regression (Appendix A). The associations remained robust when restricting full-term birth (Appendix A). When considering only toddlers with healthy mothers, we also found that the associations were similar (Appendix A).

## 4. Discussion

In this nationwide cross-sectional study in China, we found that toddlers who were not breastfed for the first six months of life had higher odds of having a diagnosis of ASD compared with those who were exclusively breastfed. We did not find a significant modification of demographic characteristics in the associations. The results remained robust in multiple sensitivity analyses.

Our findings were consistent with a recently published systematic review and meta-analysis, which reported that exclusive breastfeeding was associated with a lower risk of having ASD in children (combined OR, 0.24; 95% CI, 0.18–0.32) [32]. However, several large-scale studies indicated null associations. For example, Dodds et al. found breastfeeding at discharge was not associated with a decreased risk of ASD among 129,733 children (924 children with ASD) [14]. Soke et al. undertook a cross-sectional analysis (577 children with ASD and 794 controls, 30–68 months) from the Study to Explore Early Development in the US, which found no significant difference in breastfeeding initiation in children with ASD after adjusting for confounders [28]. Husk et al. carried out a large, nationally representative survey of US children (*n* = 37,901, 2–5 years) and found no association between breastfeeding and ASD [21]. Most of the previous studies were limited by the study area (e.g., mostly in Western countries), varied outcome definitions (e.g., parent report [13,20,21,25] or screening only [26]), and a small study sample size [11,17,18,27,30].

Data from Asian countries remained limited, and culture differences between Western and Eastern countries might have influenced the determinants of breastfeeding practices and success via culture beliefs, dietary restrictions, social support from health care providers, and so on [38]. In Asian countries, the results were more consistent. A cross-sectional study from the 2008 National Investigation of Birth Cohorts in Korea database (i.e., propensity-score-matched 188,052 children) showed that breastfeeding during the first 4 to 6 months was associated with a decreased risk of ASD compared to formula-feeding [31]. Similarly, four case-control studies (India [19,27], Indonesia [30], and Japan [12]) in Asian countries indicated a shorter duration of breastfeeding was associated with a higher risk of ASD. However, potential confounding factors (e.g., socio-economic status and autism-related risk factors) were not considered in these four studies, which might limit the accuracy of the study findings.

In China, we only found one study conducted in Shenzhen, China, which investigated the associations of breastfeeding with the risk of having autistic traits, measured via the Autism Behavior Checklist (ABC) [29]. In the current study, we further confirmed this associations based on standard screening and diagnostic processes. Furthermore, this is the first observational study with a large sample size to clarify the association between breastfeeding and the odds of ASD according to the DSM-5 criteria, and the results suggest that the associations between breastfeeding and ASD could still be observed when using the new diagnostic system.

We did not find significant modifications of demographic characteristics in the associations. Few studies have investigated potential modifications of demographic characteristics, but our results were in line with the results reported by Boucher et al., which showed no interactions between child sex and breastfeeding and autistic traits [24]. However, Brion et al.’s study [39] found that the association between breastfeeding and neural development in children can be affected by socioeconomic factors (e.g., parental education and income). Therefore, more studies are needed to investigate the potential modifications in breastfeeding–ASD associations.

Although the potential mechanism remained unclear, breastfeeding, as a uniquely close and sensual experience, could enhance social contact and create bonding between mothers and toddlers, contributing to child attachment security [16,40]. Attachment develops in accordance with the quality of mother–toddler interactions, which can promote corresponding social-emotional development and decrease the risk of ASD via positive experiences in early life [41,42]. Meanwhile, oxytocin released during breastfeeding, which is one evolutionarily conserved neuropeptide with critical functions in the control of social behaviors, especially pair-bonding (i.e., breastfeeding), could enhance social recognition and memory, and reduce stress [43,44,45]. Meta-analyses indicated that children with ASD have lower blood oxytocin levels compared to neurotypical individuals [46]. Moreover, breast milk is a rich source of micro- and macro-nutrients, such as long-chain polyunsaturated fatty acids (LCPFUAs), insulin-like growth factors (IGF) I and II, and so on, which support healthy physical growth, immune system development, and brain maturation [9,47]. LCPFUAs in breast milk, mainly omega-3 (e.g., Docosahexaenoic acid, DHA) and omega-6, might play a key role in brain development, such as the structure of the neuronal cell membranes and the maturation of the myelin sheath and retina, which are related to the development of ASD [48]. Similarly, IGF could improve myelination and promote more effective neural impulse passage [49]. These nutrients and their metabolic products have been investigated, which are involved in the etiology of ASD via multi-pathways [49,50]. In addition, poor breastfeeding practices led to deficiencies in these nutrients, disruptions in gut linings, and increased vulnerability to environmental toxins and infections [51]. Therefore, more studies are needed to confirm the role of breastfeeding in the etiology of ASD.

It should be noted that our findings have important implications for policy and practice in China. Exclusive breastfeeding up to or around 6 months is generally recommended by the World Health Organization and other international organizations [36]. Globally, 43% of the world’s infants under 6 months of age are exclusively breastfed [52]; however, only 29.2% of infants were exclusively breastfed for 6 months according to the 2017 national report in China [53]. Since countries across the world still bear substantial burdens from ASD [54], policy-led breastfeeding practices are urgently needed globally, especially in China. In addition, health care professionals and parents must be educated about the benefits of exclusive breastfeeding. Screening of toddlers during routine postnatal care to identify those without exclusive breastfeeding should be encouraged, and health promotion should also be developed to promote the awareness of breastfeeding in social communities.

There are some limitations to our study. First and foremost, the study design did not allow us to test of the causality of the associations, and the results should be interpreted with caution. Second, the breastfeeding rate in our study is quite high (48.8%), and one of the explanations for this is that our data were mostly collected in relatively developed areas with better breastfeeding practices and related services in local maternal and child health hospitals. During the study period, the Chinese government implemented national initiatives to promote breastfeeding practices by releasing breastfeeding-related systems [55]. Diverse types of evaluations, management, and supervision were used to build a friendly environment for breastfeeding, and hospitals were evaluated by the government for their baby-friendly status. The hospitals in our study were all rated as baby-friendly hospitals with better breastfeeding practices, resulting in the high prevalence of exclusive breastfeeding in this study. In addition, we did not collect the specific length of breastfeeding, and we could therefore hardly conduct dose-response analyses in this study. However, meta-analyses indicated that the associations remained similar when considering different breastfeeding durations, and this may not be a key factor to confound the associations [56]. Despite these limitations, our study has several advantages, including a large sample size, standard procedures of screening and diagnosis, and comprehensive information on potential confounders and multiple sensitivity analyses, all which strengthened our findings.

## 5. Conclusions

In summary, table our results indicated that toddlers without breastfeeding in the first six months of life had higher odds of having ASD when compared with those who were exclusively breastfed. The findings highlight the importance of strengthening public health efforts to increase exclusive breastfeeding in China, which may help to improve the health burdens of individuals with ASD.

## Figures and Tables

**Figure 1 nutrients-14-00045-f001:**
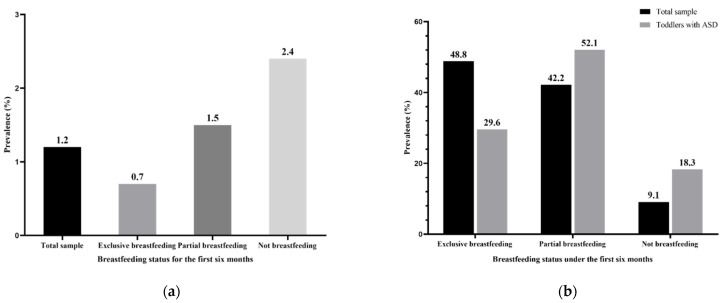
Breastfeeding status in the first six months among 6049 toddlers: (**a**) prevalence of ASD in the total sample; (**b**) prevalence of different breastfeeding statuses in total sample and toddlers with ASD.

**Table 1 nutrients-14-00045-t001:** Demographic characteristics of the 6049 toddlers in this study.

Variables		Means (Standard Deviation)/No. (%)
Child’s age (Month)		22.7 (4.1)
Child’s sex	Boy	3364 (55.6)
	Girl	2685 (44.4)
Only child	Yes	4277 (70.7)
	No	1772 (29.3)
Preterm birth	Yes	507 (8.4)
	No	5542 (91.6)
Maternal age	≤24 years	1045 (17.3)
	25–29 years	3188 (52.7)
	30–34 years	1407 (23.3)
	≥35 years	409 (6.8)
Maternal education	Primary school and below	677 (11.2)
	Middle school	1289 (21.3)
	College degree	3768 (62.3)
	Advanced degree	315 (5.2)
Yearly household income	≤¥100,000	2153 (35.6)
	¥100,000–¥300,000	2314 (38.3)
	≥¥300,000	464 (7.7)
	Not available	1118 (18.5)
Ethnic background	Han	5674 (93.8)
	Minority	375 (6.2)
Complications during pregnancy	Yes	860 (14.2)
	No	5189 (85.8)
Depression during pregnancy	Yes	1486 (24.6)
	No	4563 (75.4)
Pre-pregnancy overweight/obesity	Yes	472 (7.8)
	No	5577 (92.2)
Second-hand smoke during pregnancy	Yes	843 (13.9)
	No	5206 (86.1)

**Table 2 nutrients-14-00045-t002:** Associations between breastfeeding for the first six months of life and ASD among toddlers.

	Crude Model ^a^		Adjusted Model 1 ^b^		Adjusted Model 2 ^c^	
	OR (95% CI)	*p* Value	OR (95% CI)	*p* Value	OR (95% CI)	*p* Value
Exclusive breastfeeding	1 [Reference]		1 [Reference]		1 [Reference]	
Partial breastfeeding	2.05 (1.21, 3.57)	**0.01**	1.59 (0.93, 2.80)	0.10	1.55 (0.90, 2.74)	0.12
Not breastfeeding	3.39 (1.64, 6.73)	**<0.001**	**2.19 (1.04, 4. 46)**	**0.03**	**2.34 (1.10, 4.82)**	**0.02**

^a^ Crude model: without adjustment. ^b^ Adjusted Model 1: adjusted for child’s age, sex, only child, maternal age, maternal education level, yearly household income, ethnic background and study area. ^c^ Adjusted Model 2: further adjusted for preterm birth, pregnancy information (complications and depression during pregnancy), second-hand smoke exposure and overweight/obesity before pregnancy.

**Table 3 nutrients-14-00045-t003:** Associations of breastfeeding in the first six months with the risk of ASD stratified by demographic information ^a^.

Odd Ratios (95% CI)	Breastfeeding Status under the First Six Month			*P* _interaction_
	Exclusive Breastfeeding	Partial Breastfeeding	Not Breastfeeding	
**Children with ASD or without ASD**				
Stratified by child sex				0.66
Boy	1 [Reference]	1.42 (0.76, 2.07)	2.27 (1.39, 3.14)	
Girl	1 [Reference]	2.92 (1.30, 4.53)	2.57 (0.15, 4.99)	
Stratified by only child				0.75
No	1 [Reference]	1.16 (0.14, 2.18)	1.78 (0.38, 3.18)	
Yes	1 [Reference]	1.88 (1.12, 2.64)	2.70 (1.69, 3.72)	
Stratified by maternal age				0.45
<30 years	1 [Reference]	1.93 (1.20, 2.66)	3.03 (2.08, 3.99)	
≥30 years	1 [Reference]	0.97 (−0.15, 2.08)	1.06 (−0.58, 2.69)	
Stratified by maternal education				0.79
Middle school and below	1 [Reference]	1.27 (0.37, 2.16)	1.93 (0.86, 3.01)	
Bachelor degree or above	1 [Reference]	1.92 (1.08, 2.76)	2.84 (1.60, 4.07)	
Stratified by yearly household income				0.42
≤¥100,000	1 [Reference]	1.57 (0.74, 2.39)	1.43 (0.21, 2.65)	
>¥100,000	1 [Reference]	1.59 (0.71, 2.48)	3.73 (2.64, 4.83)	

Abbreviations: CI: confidence interval; ASD, autism spectrum disorder. ^a^ Model was adjusted for child age, sex, only child, maternal age, maternal education level, yearly household income, ethnic background, study area, preterm birth, pregnancy information (complications and depression during pregnancy), second-hand smoke exposure and overweight/obesity before pregnancy, and the differences between groups was tested by adding the interaction term.

## Data Availability

Data available on request due to restrictions eg privacy or ethical.

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
