# Peer review of "Association of Breastfeeding for the First Six Months of Life and Autism Spectrum Disorders: A National Multi-Center Study in China"

_nutrients, 2021, doi:10.3390/nu14010045_

Round 1

Reviewer 1 Report

The manuscript handles the important question on ASDs and breastfeeding.

However, while the authors suggest the lack of Asian data, there were various previous studies.

For example, in "Nutrients", (https://doi.org/10.3390/nu13082825), JH Kim suggested that breastfeeding is associated with low ASDs until 10 years.

In the current form, the article is too short and authors should clarify what has improved from each previous research. 

Also, authors are recommended to do more subgroup research to provide additional insight for readers.

Author Response

Manuscript Nutrients-1518372

Association of breastfeeding for the first six months of life and autism spectrum disorders: a national multi-center study in China

A list of the numbered detailed response is given below with the following format:

Normal: a copy of the editor and reviewer’s comment

Red: our response

Underline: copy of changed text in manuscript where appropriate

Responses to the Comments from Reviewer #1:

The manuscript handles the important question on ASDs and breastfeeding.

Response: We would like to thank the reviewer very much for the positive feedback on our manuscript and providing us with more suggestions and guidance. We have revised our manuscript to address the reviewer’s suggestions, as illustrated below. We hope that our efforts have succeeded in addressing the reviewer’s concerns, and we would be grateful for any further guidance.

Comment 1:

  • However, while the authors suggest the lack of Asian data, there were various previous studies. For example, in "Nutrients", (https://doi.org/10.3390/nu13082825), JH Kim suggested that breastfeeding is associated with low ASDs until 10 years. In the current form, the article is too short and authors should clarify what has improved from each previous research. 
  • Also, authors are recommended to do more subgroup research to provide additional insight for readers.

Response: We thank the reviewer for the comments. We have provided point-to point responses to the two comments regarding the improvement of the current study from these previous researches and subgroup analysis. Please see details below.

(1) We apologized that we did not include any discussion of the Korean study, and we have added relevant discussion in this version. Meanwhile, we have added more discussion regarding the comparisons with previous studies. Please see Page 2 Line 58-60 and Page 6-7 Line 220-242.

“One cross-sectional study from Korea showed that breastfeeding was associated with higher odds of ASD comparing to formula feeding[31].”

“…However, several large-scale studies indicated null associations. For example, Dodds et al. found breastfeeding at discharge was not associated with decreased risk of ASD among 129,733 children (924 children with ASD) [14]. Soke et al. undertook a cross-sectional analysis (577 children with ASD and 794 controls, 30-68 months) from the Study to Explore Early Development in USA which found no significant difference in breastfeeding initiation in children with ASD after adjusting confounders[28]. Husk et al. carried on a large nationally representative survey of US children (n=37,901, 2-5years) and found no association between breastfeeding and ASD [21]. Most of the previous studies were limited by the study area (e.g., mostly in Western countries), varied outcome definitions (e.g., parent report [13,20,21,25] or screening only[26]), and small study sample size[11,17,18,27,30].

Data from Asian countries remained limited, and culture difference between Western and Eastern countries might influence the determinants of breastfeeding practices and success via culture beliefs, dietary restrictions, social support from health care providers and so on [38]. In Asian countries, the results were more consistent. A cross-sectional study from the National Investigation of birth Cohort in Korea study 2008 database (i.e., propensity-score matched 188,052 children) showed that breast-feeding during the first 4 to 6 months was associated with decreased risk of ASD com-paring to formula feeding [31]. Similarly, four case-control studies (India [19,27], Indonesia [30] and Japan [12]) in Asian countries indicated shorter duration of breastfeeding was associated with higher risk of ASD. However, potential confounding factors (e.g., socio-economic status and autism-related risk factors) were not considered in these four studies, which might limit the accuracy of the study findings.”

(2) We have added subgroup analysis and provide more information about the result of subgroup analysis. Please see Page 1 Line 36-37, Page 3 Line 143-147, Page 6 Line 194-205, Page6 Line 215-216 and Page 7 Line 251-257.

“…We did not find significant modification of demographic characteristics on the associations.”

“Subgroup analysis stratified by demographic characteristics including sex, only child, maternal age, maternal education level and yearly household income was conducted, and the differences between groups was tested by adding the interaction term. We did not conduct subgroup analyses for ethnic background because the models fail to converge due to the small cases in the subgroups.”

3.4 Subgroup analysis

As shown in Table 3, there was no statistical evidence of interaction between child sex, only child, maternal age, maternal education, yearly household income and breastfeeding status on risk of ASD with P interaction = 0.662, 0.749, 0.454, 0.789 and 0.424, respectively.

Table 3 Associations of breastfeeding under the first six month with the risk of ASD stratified by demographic information a

Odd Ratios (95%CI)

Breastfeeding status under the first six month

P interaction

Exclusive breastfeeding

Partial breastfeeding

Not breastfeeding

Children with ASD or without ASD

Stratified by child sex

.66

Boy

[1] Reference

1.42 (0.76, 2.07)

2.27 (1.39, 3.14)

Girl

[1] Reference

2.92 (1.30, 4.53)

2.57 (0.15, 4.99)

Stratified by only child

.75

No

[1] Reference

1.16 (0.14, 2.18)

1.78 (0.38, 3.18)

Yes

[1] Reference

1.88 (1.12, 2.64)

2.70 (1.69, 3.72)

Stratified by maternal age

.45

< 30 years

[1] Reference

1.93 (1.20, 2.66)

3.03 (2.08, 3.99)

³ 30 years

[1] Reference

0.97 (-0.15, 2.08)

1.06 (-0.58, 2.69)

Stratified by maternal education

.79

Middle school and below

[1] Reference

1.27 (0.37, 2.16)

1.93 (0.86, 3.01)

Bachelor degree or above

[1] Reference

1.92 (1.08, 2.76)

2.84 (1.60, 4.07)

Stratified by yearly household income

.42

≤¥100, 000

[1] Reference

1.57 (0.74, 2.39)

1.43 (0.21, 2.65)

> ¥100, 000

[1] Reference

1.59 (0.71, 2.48)

3.73 (2.64, 4.83)

Abbreviations: CI: confidence interval; ASD, autism spectrum disorder.

a Model was adjusted for child age, sex, only child, maternal age, maternal education level, yearly household income, ethnic background, study area, preterm birth, pregnancy information (complications and depression during pregnancy), second-hand smoke exposure and overweight/obesity before pregnancy.

“We did not find significant modification of demographic characteristics on the associations. Few studies have investigated potential modification of demographic characteristics, but our results were in line with the results reported by Boucher et al., which showed no interaction between child sex and breastfeeding on autistic trait[24]. However, Brion et al.’s study [39] found that the association between breastfeeding and neural development in children can be affected by socioeconomic factors (e.g., parental education and income). Therefore, more studies are needed to investigate the potential modifications on the breastfeeding-ASD associations.”

Reviewer 2 Report

In the manuscript titled '" Association of breastfeeding for the first six months of life and autism spectrum disorders: a national multi-center study in China"' by Huang et al., the authors conducted a nationwide cross-sectional study in China to investigate the associations between breastfeeding custom during the first six months and the prevalence of ASD in offspring. They found that the toddlers with partial breastfeeding or without breastfeeding had higher odds of having ASD. They also performed the adjustment analyses considering demographical information and the ASD-related risk factors. The results obtained by these adjustments were similar to those in the primary finding. They further performed multiple sensitivity analyses and found that the results were still robust. The object of this study is straightforward and fits urgent medical needs. The sample number in the study is relatively large. The surveys are well-designed and appropriately analyzed. The data are clearly presented and solid. Overall, the manuscript is interesting, and the data support the main conclusions. This work clearly demonstrates the importance of breastfeeding custom to prevent ASD. This work appears to be worth publishing in Nutrients. However, a few minor but critical issues need to be addressed and clarified before publication.

Major points

  1. The authors refer to 'Figure 1' (P.4 Line 164) in the manuscript, although the figure is missing in the current manuscript. And they show 'Figure 2', but they do not mention Figure 2 in the manuscript. They might have mislabeled Figure 1 as Figure 2. They should amend it.
  2. They described in the sentence as follows 'Among 6,049 toddlers, 71 toddlers were diagnosed with ASD with a prevalence rate 163 of 1.2%, and more boys (n=62) were identified as ASD (Figure 1)' (P. 4 Lines 163-164). It looks awkward that this primary result (the prevalence rate of ASD is 1.2%) is shown in Fig. 1B. This order of description should coincide with the order of the figures. Because this is the primary finding in the research, the authors should show the data (the prevalence rate of ASD is 1.2%) in Fig. 1A, provided that Figure 1 is mislabeled as Figure2. Then they should refer to their key finding that the prevalence rate of ASD among exclusive breastfeeding, partial breastfeeding, and none breastfeeding is 48.8%, 42.2%, and 9.1%, respectively, which should be shown as Fig. 1B.

Author Response

Manuscript Nutrients-1518372

Association of breastfeeding for the first six months of life and autism spectrum disorders: a national multi-center study in China

A list of the numbered detailed response is given below with the following format:

Normal: a copy of the editor and reviewer’s comment

Red: our response

Underline: copy of changed text in manuscript where appropriate

Responses to the Comments from Reviewer #2:

In the manuscript titled '" Association of breastfeeding for the first six months of life and autism spectrum disorders: a national multi-center study in China"' by Huang et al., the authors conducted a nationwide cross-sectional study in China to investigate the associations between breastfeeding custom during the first six months and the prevalence of ASD in offspring. They found that the toddlers with partial breastfeeding or without breastfeeding had higher odds of having ASD. They also performed the adjustment analyses considering demographical information and the ASD-related risk factors. The results obtained by these adjustments were similar to those in the primary finding. They further performed multiple sensitivity analyses and found that the results were still robust. The object of this study is straightforward and fits urgent medical needs. The sample number in the study is relatively large. The surveys are well-designed and appropriately analyzed. The data are clearly presented and solid. Overall, the manuscript is interesting, and the data support the main conclusions. This work clearly demonstrates the importance of breastfeeding custom to prevent ASD. This work appears to be worth publishing in Nutrients.

Response: We would like to thank the reviewer very much for the positive feedback on our manuscript and providing us with more suggestions and guidance. We have revised our manuscript to address the reviewer’s suggestions, as illustrated below. We hope that our efforts have succeeded in addressing the reviewer’s concerns, and we would be grateful for any further guidance.

Comment 1

However, a few minor but critical issues need to be addressed and clarified before publication.

  • The authors refer to 'Figure 1' (P.4 Line 164) in the manuscript, although the figure is missing in the current manuscript. And they show 'Figure 2', but they do not mention Figure 2 in the manuscript. They might have mislabeled Figure 1 as Figure 2. They should amend it.
  • They described in the sentence as follows 'Among 6,049 toddlers, 71 toddlers were diagnosed with ASD with a prevalence rate 163 of 1.2%, and more boys (n=62) were identified as ASD (Figure 1)' (P. 4 Lines 163-164). It looks awkward that this primary result (the prevalence rate of ASD is 1.2%) is shown in Fig. 1B. This order of description should coincide with the order of the figures. Because this is the primary finding in the research, the authors should show the data (the prevalence rate of ASD is 1.2%) in Fig. 1A, provided that Figure 1 is mislabeled as Figure2. Then they should refer to their key finding that the prevalence rate of ASD among exclusive breastfeeding, partial breastfeeding, and none breastfeeding is 48.8%, 42.2%, and 9.1%, respectively, which should be shown as Fig. 1B.

Response: We thank the reviewer for the comments. We have provided point-to point responses to the two comments regarding the epidemiological methods. Please see details below.

  • We apologized for our mistakes, and we have revised it accordingly. Please see Page 5 Line 177 and Figure 1.

Figure 1. Breastfeeding status under the first six months among 6,049 toddlers:”

  • We have revised the order of the figure according to the order of result. Please see Figure 1.

(a)

(b)

Figure 1. Breastfeeding status under the first six months among 6,049 toddlers: (a) prevalence of ASD in total sample; (b) prevalence of different breastfeeding status in total sample and toddlers with ASD.

Round 2

Reviewer 1 Report

Ln 58: maybe reverted?

Otherwise the manuscript is acceptable in present form.